# Face Mask Wearing Detection Algorithm Based on Improved YOLO-v4

**DOI:** 10.3390/s21093263

**Published:** 2021-05-08

**Authors:** Jimin Yu, Wei Zhang

**Affiliations:** 1College of Automation, Chongqing University of Post and Telecommunications, Chongqing 400065, China; yujm@cqupt.edu.cn; 2Key Lab of Industrial Wireless Networks and Networked Control of the Ministry of Education, Chongqing 400065, China

**Keywords:** adaptive image scaling, CSPDarknNet53, face mask recognition, PANet, YOLO-v4

## Abstract

To solve the problems of low accuracy, low real-time performance, poor robustness and others caused by the complex environment, this paper proposes a face mask recognition and standard wear detection algorithm based on the improved YOLO-v4. Firstly, an improved CSPDarkNet53 is introduced into the trunk feature extraction network, which reduces the computing cost of the network and improves the learning ability of the model. Secondly, the adaptive image scaling algorithm can reduce computation and redundancy effectively. Thirdly, the improved PANet structure is introduced so that the network has more semantic information in the feature layer. At last, a face mask detection data set is made according to the standard wearing of masks. Based on the object detection algorithm of deep learning, a variety of evaluation indexes are compared to evaluate the effectiveness of the model. The results of the comparations show that the mAP of face mask recognition can reach 98.3% and the frame rate is high at 54.57 FPS, which are more accurate compared with the exiting algorithm.

## 1. Introduction

Things such as respiratory infection viruses, toxic and harmful gases and dust suspended in the air can enter the lungs of humans as they breathe and then cause pneumonia, nerve damage and toxic reactions. In particular, the new coronavirus (COVID-2019) has spread globally since the end of 2019 which has a great impact on the safety of lives and property of all human beings. When people are exposed to toxic or harmful gases, wearing masks can effectively protect them from being endangered, thereby reducing unnecessary losses [1]. Therefore, it is of great practical significance to realize the mask wearing detection algorithm.

At present, in places where masks need to be worn (such as communities, campuses, supermarkets, hospitals, factories, stations, etc.), the wearing of masks is usually checked manually. However, this method would cause a waste of human resources and low efficiency, and most importantly, there are problems such as missing and false detection. Object detection technology enables us to use the camera and computer integrated way to realize the face mask wearing detection so that the purpose of non-contact automatic detection is achieved.

In ref. [2], the authors proposed the LeNet-5 network architecture, which is a classic work of the convolutional neural network and provides great help for the development of computer vision. However, due to the influence of computing power and a lack of data sets at that time, the neural network model in ref. [2] is surpassed by the effect of SVM [3] under certain computing power conditions. Based on deep learning, until 2012, Hinton [4] firstly proposed the AlexNet convolutional neural network model, which is respected as a solid foundation for the development of an object detection algorithm. In that work, the author used the ReLU activation function [5] to speed up the training of the network during the gradient descent process of the model and introduced a Dropout layer to suppress over-fitting [6] so that the network can extract object features more effectively. In 2014, He [7] extracted object feature in areas with any aspect ratio by using a Spatial Pyramid Pooling Network (SPPNet) method, which provided ideas for YOLO-v3 [8], YOLO-v4 [9] and other detection algorithms to extract features at any scale. In the next year, the residual block structure in ResNet was introduced to improve the feature expression ability of the model in [10]. Based on Feature Pyramid Networks (FPN) [11], Liu [12] put forward the Path Aggregation Network (PANet) to prove the importance of the underlying information of the feature layer in 2018, thus realizing the circulation of object feature information.

At present, object detection algorithms based on deep learning are usually divided into two categories. The first is the Two-Stage algorithm based on the R-CNN [13,14,15] and TridenNet [16], etc. The existing problems of such Two-Stage algorithms are poor real-time, large model scale and poor small object detection effect. The second is the One-Stage algorithm based on the SSD [17,18,19,20] and YOLO [21,22], which has high real-time performance in multi-scale object detection. However, the detection accuracy needs to be improved.

Combined with the advantages of YOLO series object detection algorithms, some improved methods of CSPDarkNet53 and PANet are introduced into YOLO-v4 in the present paper and a model which can enable the mask detection task and achieve optimal performance is developed. Similarly to papers [23,24,25], we build the network model based on deep learning and computer-aided diagnosis. The method used in this article is described in the flowing chart in Figure 1.

The contributions of this paper are as follows:Aiming at the problem of training time, this paper introduces the improved CSPDarkNet53 into the backbone to realize the rapid convergence of the model and reduce the time cost in training.An adaptive image scaling algorithm is introduced to reduce the use of redundant information in the model.To strengthen the fusion of multi-scale semantic information, the improved PANet is added into the Neck module.The Hard-Swish activation function introduced in this paper can not only strengthen the nonlinear feature extraction ability of the network, but also enable the detection results of the model to be more accurate.

To sum up, in the face mask detection task, the algorithm proposed in this paper has higher detection accuracy than other typical algorithms, which means the algorithm is more suitable for the mask detection task. At the same time, the algorithm is more practical to deploy in public places to urge people to wear masks regularly in order to reduce the risk of cross-infection.

## 2. Related Works

### 2.1. Problems Exist in Object Detection

There are two key points of face mask wearing detection. One is to locate the position of the face in the image; the other is to identify whether the face given by the data set is wearing a mask and if the mask is worn correctly. Problems of the present object detection algorithm can be attributed to face occlusion, variable face scale, uneven illumination, density, etc., and these problems seriously affect the performance of the algorithm. Furthermore, the traditional object detection algorithm adopts the selective search method [26] in feature extraction, leading to problems such as poor generalization ability, redundant information, low accuracy and poor real-time performance.

### 2.2. Existing Work

Some researchers have used the extraction of RGB color information to perform face mask recognition [27]. However, the article does not consider the case of non-standard wearing of masks, so the adaptability of the algorithm needs to be further improved. Combining YOLO-v2 and ResNet50, the authors in [28] realized face mask recognition whose backbone network is DarkNet-19. However, DarkNet-19 has been optimized by CSPDarkNet53. The ablation experiment in our paper shows that the CSP1_X module produces better results than CSPDarkNet53. In [29], the authors pointed out that the combination of ResNet50 and SVM can realize face mask detection and its accuracy can reach up to 99.64%. However, the algorithm takes a lot of computational costs. Furthermore, the combination of SSD and MobileNetV2 for mask detection was proposed in paper [30], but its model structure is too complex and its performance is inferior to YOLO-v4.

Only two categories are used in the papers mentioned in the above paragraph and the authors did not consider the influence of wearing masks irregularly on the algorithm. Therefore, the feature extraction ability and model practicability of these algorithms need to be improved. In this paper, based on improved YOLO-v4, face mask recognition is considered and three categories, face_mask, face and WMI, are included. In addition, the feature extraction ability of this paper is improved by CSP1_X, and CSP2_X impels PANet to speed up the circulation of semantic features and strengthen feature fusion, thus improving the robustness of the model.

## 3. The Model Structure of YOLO-v4 Network

YOLO-v4 is a high-precision and real-time One-Stage object detection algorithm based on regression proposed in 2020, which integrated the characteristics of YOLO-v1, YOLO-v2, YOLO-v3, etc., and achieved the current optimum in terms of detection speed and trade-off of detection accuracy. The model structure is shown in Figure 2, which consists of three parts: Backbone, Neck, and Prediction.

Combined with the characteristics of the ResNet structure, YOLO-v3 integrated the residual module into itself and then obtained Darknet53. Based on this, taking the superior learning ability of Cross-Stage Partial Network (CSPNet) [31] into account, YOLO-v4 constructed the CSPDarkNet53. In the residual module, the feature layer is input and the higher-level feature information is output. This means the learning goal of the model in the ResNet module becomes the difference between the output and the input, thus realizing residual learning while reducing the parameters of the model and strengthening feature learning. The Neck can be composed of the SPPNet and PANet. In SPPNet, firstly, the feature layer is convolved three times, and then the input feature layer is maximally pooled by using the maximum pooling cores of different sizes. The pooled results are concatenated firstly and then convolved three times, thus improving the network receptive field. PANet convolves the feature layers after the operation of Backbone and SPPNet and then up-samples them, that is, making the original feature layers double in height and width, and then concatenates the feature layers after convolution and up-sampling with the feature layers obtained by CSPDarkNet53 to realize feature fusion, and then down-sampling, compressing the height and width, and finally stacking with the previous feature layers to realize more feature fusion (five times). The Prediction module can make predictions by using the feature extracted from the network. Taking a 13 × 13 grid, for example, it is equivalent to divide the input picture into 13 × 13 grids, and then each grid will be preset with three prior frames. The prediction results of the network will adjust the positions of the three prior frames, and finally, it will be filtered by the non-maximum suppression (NMS) [32] algorithm to obtain the final prediction frame.

YOLO-v4 proposed a new mosaic data augmentation method to expand the data set and introduced CIOU as the positioning loss function [33], which made the network more inclined to optimize in the direction of increasing overlapping areas, thus effectively improving the accuracy. In the actual complex environment, due to the external interference such as occlusion and multi-scale, there are still some shortcomings in the face mask detection directly using YOLO-v4. The main performances are as follows:

There are still problems such as insufficient shallow feature extraction for multi-scale objects. 

In the reasoning stage, the model adds gray bars at both ends of the image to prevent the image from distorting, but too many gray bars increase the redundant information of the model. 

At the same time, the model has problems such as long training time, high calculation cost and overfull parameters. 

To solve these problems, this paper optimizes and improves the model based on YOLO-v4.

## 4. Improved YOLO-v4 Network Model

With the increasing number of layers of the convolutional neural network, the depth of the network is deepening, and the deeper network structure is beneficial for the extraction of object features. Thereupon, the semantic information of small objects is increased. The main improvements presented in this paper based on YOLO-v4 are as follows: The CSPDarkNet53 is improved into CSP1_X and CSP2_X, and so reduced network modules to reduce the parameters of feature extraction in the network model; using the CSP2_X module in Neck can increase information fusion, and the adaptive image scaling method is used to replace the image scaling method in YOLO-v4.

### 4.1. Backbone Feature Extraction Network

The residual module introduced into YOLO-v4 is to enhance the learning ability of the network and reduce the number of parameters. The operation process of the residual module (Res-unit) can be summed up as follows. Firstly, perform 1 × 1 convolution; then 3 × 3 convolution; and weighting the two outputs of the module at last. The purpose of weighting is to increase the information of the feature layer without changing its dimension information. In CSPDarkNet53, the set of feature layers of the image is input, and then convolution down-sampling is performed continuously to gain higher semantic information. Therefore, the last three layers of Backbone have the highest semantic information, and then the last three layers of features are selected as the input of SPPNet and PANet. The network structure of CSPDarkNet53 is shown in Figure 3.

Although YOLO-v4 uses the residual network to reduce the computing power requirement of the model, its memory requirement still needs to be improved. Therefore, in this paper, the network structure of CSPDarkNet53 of YOLO-v4 is improved to the CSP1_X module, as shown in Figure 4.

Compared with CSPDarkNet53 in Figure 2, the improved network uses the H-swish activation function [34], as shown in Equation (1):(1)H−swish(x)=xReLU(x+3)6

As the Swish function [35] contains the Sigmoid function, the calculation cost of the Swish function is higher than the Re*LU* function, but the Swish function is more effective than the Re*LU* one. Howard used the H-swish function on mobile devices [36] to reduce the number of accesses to memory by the model, which further reduced the time cost. Therefore, in this paper, the advantages of the H-swish function are used to reduce the running time requirements of the model on condition of ensuring no gradient explosion, disappearance and other problems. At the same time, the detection accuracy of the model is advanced.

In CSP1_X, the input feature layer of the residual block is divided into two branches. One is used as the residual edge for convolution operation. The other plays the role of the trunk part, performs 1 × 1 convolution operation at first, then performs 1 × 1 convolution to adjust the channel after entering the residual block, and then performs the 3 × 3 convolution operation to enhance the feature extraction. At last, the two branches are concatenated, thus merging the channels to obtain more feature layer information. In this paper, three CSP1_X modules are used in the improved Backbone, where X represents the number of residual weighting operations in the residual structure. Finally, after stacking, a 1 × 1 convolution is used to integrate the channels. Experiments show that using this residual structure can make the network structure easier to optimize.

### 4.2. Neck Network

The convolutional neural network requires the input image to have a fixed size. In the past convolution neural network, the fixed input was obtained by cutting and warping operations, but these methods easily bring about problems such as object missing or deformation. To eliminate such problems, researchers proposed SPPNet to remove the requirement of fixed input size. To gain multi-scale local features, YOLO-v4 introduced the SPPNet structure based on YOLO-v3. In order to further fuse the multi-scale local feature information with the global feature information, we add the CSP2_X module to the PANet structure of YOLO-v4 to enhance the feature extraction, which helps to speed up the flow of feature information and enhance the accuracy of the model. CSP2_X is shown in Figure 5.

The common convolution operation is adopted in the Neck network in YOLO-v4, while the CSPNet has the advantages of superior learning ability, reduced computing bottleneck and memory cost. Adding the improved CSPNet network module based on YOLO-v4 can further enhance the ability of network feature fusion. This combined operation can realize the top-down transmission of deeper semantic features in PANet, and at the same time fuse the bottom-up deep positioning features from the SPPNet network, thus realizing feature fusion between different backbone layers and different detection layers in the Neck network and providing more useful features for the Prediction network.

### 4.3. Adaptive Image Scaling

In the object detection network, the image data received by the input port have a uniform standard size. For example, the standard size of each image in the handwritten numeral recognition data set MNIST is 28 × 28. However, different data sets have different image sizes, and ResNet fixes the input image to 224 × 224. Two standard sizes of 416 × 416 and 608 × 608 are provided at the input port of the YOLO-v4 detection network. Traditional methods for obtaining standard size mainly include cutting, twisting, stretching, scaling, etc., but these methods are easy to cause problems such as missing objects, loss of resolution and a reduction in accuracy. In the previous convolutional neural network, it was necessary to unify the image data to the standard size manually in advance, while YOLO-v4 standardized the image size directly by using the data generator, and then input the image into the network to realize the end-to-end learning process. In the training and testing stage of YOLO-v4, the sizes of input images are 416 × 416 and 608 × 608. When standardizing images of different sizes, the original images are scaled firstly; then gray images with sizes of 416 × 416 or 608 × 608 are generated; and finally, the scaled image is overlaid on the gray image to obtain image data of standard size.

Taking the unregulated mask image as an example, the image processed by the scaling algorithm is shown in Figure 6. In Figure 6, A is the original input image. After calculating the scaling ratio of the original input image, image B is obtained by the BICUBIC interpolation algorithm [37], and C is a gray image with the standardized size. Finally, the original image can be pasted into the gray image, and the standardized standard input image D can be obtained. The image scaling algorithm reduces the resolution without distortion, but in practical applications, most images have different aspect ratios, so after scaling and filling by this algorithm, the gray image sizes at both ends of the image are not the same. If the filled gray image size is too large, there is information redundancy which increases the reasoning time of the model.

Therefore, this paper introduces an adaptive image scaling algorithm to adaptively add the least red edge to the original image. The steps of this algorithm are shown in Algorithm 1.


**Algorithm 1** Adaptive image scaling.Input: W and H are the width and height of the input image.$\;T_{W}$ and $T_{H}$ are the width and height of the object image of standard size.Begin$\;scaling\_ratio\leftarrow \min\{T_{W}/W,T_{H}/H\}$         new_w←W×scaling_ratio         new_h←H×scaling_ratio $\;d_{w}\leftarrow T_{W}-new\_w$ $\;d_{h}\leftarrow T_{H}-new\_h$ $\;d\leftarrow \mathrm{mod}(\max(d_{w},d_{h}),64)$         padding←d/2          if (W,H)≠(neww_,new_h):
            image←resize(input_image,(new_w,new_h))         new_image←add_border(image,(padding,padding))
EndOutput: new_image



The process of this algorithm can be understood as follows: In the first step, $T_{W}$ and $T_{H}$ in the standard size of the object image are divided by W and H of the input image, respectively, and then their minimum value scaling_ratio is treated as the scaling factor. In the second step, multiply the scaling factor by the original image’s W, H, respectively, and then take new_w and new_h as the scaled dimensions of the original image. In the third step, the $T_{W}$ and $T_{H}$ of the object image are subtracted by new_w and new_h, respectively, to obtain $d_{w}$ and $d_{h}$. In the fifth step, obtain the maximum value of $d_{w}$ and $d_{h}$, and then calculate the remainder of this maximum value and 64. As the network model will carry out five down-sampling operations, the size of the original input image is five times that of the feature graphs after five down-sampling operations, and each down-sampling operation can compress the height and width of the feature graphs of the last time to ½ of the original. Therefore, the size of the feature map obtained after five down-sampling operations is 1/32 of the original image, so the length and width must be multiples of 32. In this paper, 64 is also required, and the remainder is assigned as d. The fifth step is to calculate the padding of the red edge on both sides of the image. For the sixth step, if W, H and new_w, new_h are not the same, scale the original image to the image of new_w and new_h, respectively. The last step is to fill the two sides of the image after scaling to obtain a new image.

Similarly, the image of wearing a mask irregularly, for instance, after being processed by the adaptive scaling algorithm, is shown in Figure 7.

Comparing Figure 6 with Figure 7, it can be observed that the original image adds the least red edge at both ends of the image after adaptive scaling, thus reducing redundant information. When the model is used for reasoning, the calculation will be reduced, and the reasoning speed of object detection will be promoted.

### 4.4. Improved Network Model Structure

The improved network model is shown in Figure 8, in which three CSP1_X modules are used in the Backbone of the backbone feature extraction network, and each CSP1_X module has X residual units. In this paper, considering the calculation cost, the residual modules are connected in series into the combination of X residual units. This operation can replace the two 3 × 3 convolution operations with a combination of 1 × 1 + 3 × 3 + 1 × 1 convolution module. The first 1 × 1 convolution layer can compress the number of channels to half of the original one and reduce the number of parameters at the same time. The 3 × 3 convolution layer can enhance feature extraction and restore the number of channels. The last 1 × 1 convolution operation restores the output of the 3 × 3 convolution layer, so the alternate convolution operation is helpful for feature extraction, ensures accuracy and reduces the amount of computation.

The Neck network is mainly composed of the SPPNet and improved PANet. In this paper, the SPPNet module enlarges the acceptance range of backbone features effectively, and thus significantly separates the most important contextual features. The high computational cost of model reasoning is mainly caused by the repeated appearances of gradient information in the process of network optimization. Therefore, from the point of view of network model design, this paper introduces the CSP2_X module into PANet to divide the basic feature layer from Backbone into two parts and then reduces the use of repeated gradient information through cross-stage operation. In the same way, the CSP2_X module uses the combination of 1 × 1 + 3 × 3 + 1 × 1 convolution module to reduce the computation cost and ensure accuracy.

The Prediction module uses the features extracted from the model to predict. In this paper, the Prediction network is divided into three effective feature layers: 13 × 13 × 24, 26 × 26 × 24 and 52 × 52 × 24, which correspond to big object, medium object and small object, respectively. Here, 24 can be understood as the product of 3 and 8, and 8 can be divided into the sum of 4, 1 and 3, where 4 represents the four position parameters of the prediction box, 1 is used to judge whether the prior box contains objects, and 3 represents that there are three categories of mask detection tasks.

### 4.5. Object Location and Prediction Process

The YOLO-v3, YOLO-v4 and the models used in this paper are all predicted by the Prediction module after extracting three feature layers. For the 13 × 13 × 24 effective feature layer, it is equivalent to divide the input picture into 13 × 13 grids, and each grid will be responsible for object detection in the area corresponding to this grid. When the center of an object falls in this area, it is necessary to use this grid to take charge of the object detection. Each grid will preset three prior boxes, and the prediction results of the network will adjust the position parameters of the three prior boxes to obtain the final prediction results. Similarly, the prediction process of effective feature layers of 26 × 26 × 24 and 52 × 52 × 24 is the same as that of feature layers of 13 × 13 × 24.

In Figure 9, the feature layer is divided into 13 × 13 grids to illustrate the process of object location and prediction. Figure 9a represents the original input image of three-channel color with a size of 416 × 416 × 3. Figure 9b is obtained from the feature extraction of the input image through the network, which represents the effective feature layer with the size of 13 × 13 × 24 in the Prediction module. The feature layer is divided into 13 × 13 grids and each grid has three prior boxes which are represented by green boxes. Their center points are $c_{x}$ and $c_{y}$, width and height are $p_{w}$ and $p_{h}$, respectively. The final prediction box is a blue box with center points $t_{x}$ and $t_{y}$, width and height $b_{w}$ and $b_{h}$, respectively. Figure 9c is an input image mapped by Figure 9b, which means that the size of the prior box, grid point, prediction box, height and width in Figure 9c is 32 times that of Figure 9b. Therefore, when the center of the face wearing a mask irregularly falls within the orange box, this grid is responsible for face detection. The prediction results of the network will adjust the positions of the three prior boxes, and then the final prediction box will be screened out by ranking the confidence level and NMS to obtain Figure 9d as the detection result of the network.

YOLO-v3 is an improved version based on YOLO-v2, which solves the multi-scale problem of objects and improves the detection effect of the network on small-scale objects. At the same time, YOLO-v3 uses binary cross-entropy as the loss function, so that the network can realize multi-category prediction with one boundary box. YOLO-v3 and YOLO-v4 prediction methods are adopted in the prediction process in the present paper, as shown in Figure 9b, and $t_{x}$, $t_{y}$, $t_{w}$ and $t_{h}$ are the four parameters that the network needs to learn, which are:(2)$$b_{x}=\sigma (t_{x})+c_{x}$$
(3)$$b_{y}=\sigma (t_{y})+c_{y}$$
(4)$$b_{w}=p_{w}e^{t_{w}}$$
(5)$$b_{h}=p_{h}e^{t_{h}}$$

In the training process, the network constantly learns four parameters $t_{x}$, $t_{x}$, $t_{y}$ and $t_{w}$, thus constantly adjusting the position of the prior box to approach the position of the prediction box, and finally obtaining the final prediction result. $\sigma (t_{x})$ and $\sigma (t_{y})$, respectively, represent that $t_{x}$ and $t_{y}$ are constrained by the Sigmoid function to ensure that the center of the prediction box falls within the grid.

The confidence score reflects the accuracy of the model predicting that an object is a certain category, as shown in Equation (6).
(6)$$Confidence=P_{r}\left(Class_{i}|Object\right)\times P_{r}\left(Object\right)\times IoU_{pred}^{truth}$$

In Equation (6), $P_{r}\left(Class_{i}|Object\right)$ means the probability of what kind of object it is when it is known to be an object. $P_{r}\left(Object\right)$ represents the probability of whether the prediction box contains an object. If an object is included, $P_{r}\left(Object\right)$ = 1, otherwise it equals 0. $IoU_{pred}^{truth}$ tells us the overlap ratio between the predicted box and the true box [38].

### 4.6. The Size Design of Prior Box

For the mask detection data set in this paper, it is necessary to set appropriate prior box sizes to obtain accurate prediction results. The size of the prior box obtained by the k-means clustering algorithm is shown in Table 1.

## 5. Experimental Data Set

### 5.1. Data Set

At present, the published mask data sets are few, and there are problems such as poor content, poor quality and single background which cannot be directly applied to the face mask detection task in a complex environment. Under such context, this paper adopts the method of using my own photos and screening from the published RMFD [39] and MaskedFace-Net [40] data sets to manufacture a data set of 10,855 images, of which 7826 are selected for training, 868 for verification and 2161 for testing. When creating the data set, we fully consider the mask type, manufacturer, color and other factors to meet the richness of the data set. Therefore, the model algorithm has stronger generalization ability and detection ability in practical use. People’s behavior of covering their faces with objects that are not masks easily leads to the false detection of objects in the algorithm, hence, we treat this kind of behavior as "face". In the whole data set, there are 3615 images without masks, 3620 images with masks regularly and 3620 images with masks irregularly. The face in each picture corresponds to a label, and each label corresponds to a serial number. In this paper, the detection tasks are divided into three categories: serial number 0 corresponds to the “face”, indicating that no mask is worn; serial number 1 is equal to “face_mask”, showing that the face wears a mask regularly; and serial number 2 is equivalent to “WMI”, which means wearing masks irregularly. The sample distribution of different categories in the data set is shown in Table 2, where images represent the number of categories and objects represent the number of instances [41].

### 5.2. Region Division of Real Box

Whether a face is standard for wearing a mask can be judged by the exposure of the nose, mouth and chin in the face. We randomly selected 100 original images from the data set as the research object, and the area between eyebrows and chin in the images as the research area. We can conclude that the nose is located at the height of 28.5–55% of the image. The mouth is distributed in 55–81% of the image. The chin is located at 81–98% of the image, as shown in Figure 10. Therefore, based on this conclusion, we use the LabelImg tool to label every face in each picture in the data set and determine its category and coordinate information to obtain the real box.

Generally speaking, the face is completely exposed if not wearing a mask. The nonstandard wearing of masks can be attributed to four situations: the nose is exposed; the nose and mouth are exposed; the mouth, nose and chin are all exposed; and only the chin is exposed. To wear a mask in a standard way, you need to ensure that the front and back of the mask are correctly distinguished, and the upper and lower sides of the mask must be used to press the metal strips on both sides of the nose bridge with both hands to make the upper end of the mask close to the bridge of the nose, and the left and right ends of the mask close to the cheeks. Then, stretch the mask downwards so that the mask does not leave wrinkles and better covers the nose, mouth and chin. Figure 11 shows the standard and non-standard way of wearing a mask.

## 6. Experimental Results and Analysis

To verify the advantages of the improved model compared with other detection models, a great deal of experiments are carried out to illustrate the validity of the model performance.

### 6.1. Experimental Platform and Parameters

The configuration parameters of the software and hardware platform implemented by the algorithm in this paper are shown in Table 3.

Before the model in this article starts to be trained, its hyperparameters need to be initialized. The model continuously optimizes the parameters during the training process so that it speeds up the convergence of the network and prevents it from overfitting. All experiments in this paper are performed under the epoch of 50, batch size of 8 and the input image size of 416 × 416 × 3. The parameter adjustment process is shown in Table 4.

### 6.2. The Performance of Different Models in Training

In the training process, the model updates its parameters from the training set to achieve better performance. To verify the effect of CSP1_X and CSP2_X modules on the improved model, this paper compares the training performance with other object detection models, as shown in Table 5.

It can be seen that with the parameters of this model, 15.9 MB are reduced compared with YOLO-v4 and 13.5 MB are less than YOLO-v3. At the same time, the model size is 0.371 and 0.387 times that of YOLO-v4 and YOLO-v3, respectively. Under the same conditions, the training time of this model is 2.834 h, which is the lowest of all the models compared in the experiment.

In Faster R-CNN, the authors used the Region Proposal Network (RPN) to generate W × H × K candidate regions, which increases the operation cost. Meanwhile, Faster R-CNN not only retained the ROI-Pooling layer in it but also used the full connection layer in the ROI-Pooling layer, which brought the network many repeated operations and then reduced the training speed of the model.

### 6.3. Comparision of Reasioning Time and Real-Time Performance

In this paper, video plays a role to verify the real-time performance of the algorithm. FPS (Frames Per Second) is often used to characterize the real-time performance of the model. The larger the FPS becomes, the better the real-time performance will be. In the meantime, the adaptive image scaling method is used to verify the reliability of the algorithm in the reasoning stage, as shown in Table 6.

In the present work, we use the same picture to calculate the test time and compare the total reasoning time on the test set. It can be seen from the table that the adaptive image scaling algorithm can effectively reduce the size of red edges at both ends of the image, and the detection time consumed in the reasoning process is 144.7 s, which is 6.4 s less than that of YOLO-v4. However, thanks to its model structure, SSD consumes the shortest reasoning time. Faster R-CNN consumes the most time in reasoning, which is a common feature of the Two-Stage algorithm. Meanwhile, the FPS of our algorithm can reach 54.57 FPS, which is the highest among all comparison algorithms, while Faster R-CNN reaches the lowest.

### 6.4. The Parameter Distuibution of Different Network Layers

In general, the spatial complexity of the model can be reflected by the total number of parameters. We analyze the distribution of parameters from various network parts of different models in this paper, thus verifying the effectiveness of the improved backbone feature extraction network and PANet, as shown in Table 7.

In YOLO-v4, its parameters are mainly distributed in the backbone feature extraction network, and a different number of residual modules is used to extract deeper information, but as the network gets deeper, the parameters will become more and this will complicate the model. It can be seen from Table 7 that the algorithm in this paper has fewer parameters in the backbone network, which is due to the use of the shallower CSP1_X module, and it effectively reduces the size of the model. Furthermore, five CSP2_X modules are used in the Neck module to gather more parameters, which is more helpful to enhance feature fusion. At last, our model has 335 layers in total, 35 less than YOLO-v4.

### 6.5. Model Testing

After the model training is completed, the trained weights are used to test the model, and the model is evaluated from many aspects. For our face mask data set, the test results can be classified into three categories: TP (true positive) means that the categories in the test set are the same as the test results; FP (false positive) means the number of samples in the detected object category is inconsistent with the real object category; and FN (false negative) indicates that the real sample is detected as the opposite result or in the undetected category. For all positive cases judged by the model, the number is (TP+FP), so the proportion of real cases (TP) is called the precision rate, which represents the proportion of samples of real cases in positive cases among samples detected by the model, as shown in Equation (7).
(7)$$Precision=\frac{TP}{TP+FP}$$

For all positive examples in the test set, the number is (TP+FN). Therefore, the recall rate is used to measure the ability of the model to detect the real cases in the test set, as shown in Equation (8).
(8)Recall=TPTP+FN

To characterize the precision of the model, this article introduces AP (Average Precision) and mAP (mean Average Precision) indicators to evaluate the accuracy of the model, as shown in Equations (9) and (10).
(9)$$AP=\int_{0}^{1}P(R)dR$$
(10)$$mAP=\frac{\sum_{i=1}^{N}AP_{i}}{N}$$

Among them, P,R,N, respectively, represent precision, recall rate and the total number of objects in all categories.

Through Equations (7) and (8), it can be found that there is a contradiction between precision rate and recall rate. Therefore, the comprehensive evaluation index F-Measure used to evaluate the detection ability of the model can be shown as:(11)$$F_{\alpha }=\frac{(\alpha^{2}+1)\times P\times R}{\alpha^{2}(P+R)}$$

When α = 1, $F_{1}$ represents the harmonic average of precision rate and recall rate, as shown in Equation (12):(12)$$F_{1}=\frac{2\times P\times R}{\left(P+R\right)}$$

If $F_{1}$ is higher, the test of the model will be more effective. We use 2161 images with a total of 2213 objects as the test set. The test results of the model with IOU = 0.5 are shown in Table 8.

It can be seen from Table 8 that the model in this paper reaches the maximum value in TP and the minimum value in FN, and this means that the model itself has good detection ability for samples. At the same time, the model reaches the optimal value in the $F_{1}$ index compared with other models.

To further compare the detection effect of our model with YOLO-v4 and YOLO-v3 on each category in the test set, the AP value comparison experiments of several models are carried out under the same experimental environment, as shown in Table 9.

It can be seen from Table 9 that the performance of the AP value of our model is higher than YOLO-v4, YOLO-v3, SSD and Faster R-CNN under different IOUs, thus the average precision of our model is effectively verified. However, in the case of SSD in AP@.50, its AP in the category of “WMI” reaches the highest value.

In this paper, mAP is introduced to measure the detection ability of the model for all categories, and the model is tested on IOU = 0.5, IOU = 0.75 and IOU = 0.5:0.05:0.95 to further evaluate the comprehensive detection ability of the model, as shown in Table 10.

It can be seen that when IOU = 0.5, the mAP of this model is 3.1% higher than that of YOLO-v4 and 1.1% higher than SSD. Under the condition of IOU = 0.5:0.95, a more rigorous test is carried out, and the experiment shows that mAP@.50:95 is 16.7% and 15.6% higher than YOLO-v4 and SSD, respectively. This fully shows that the model is superior to YOLO-v4 and SSD in comprehensive performance. It is worth pointing out that the mAP of Faster R-CNN is higher than YOLO-v4 and YOLO-v3, but the FPS is the lowest, which also implies the common characteristics of the Two-Stage detection algorithm: high detection accuracy and low real-time. At the same time, we illustrate the performance of different models in terms of test performance in a visual way, as shown in Figure 12.

The pictures used in the comparative experiment in Figure 12 are from the test set of this paper. Each experiment is conducted in the same environment. Meanwhile, visual analysis is carried out on condition of the confidence level 0.5. In the figure, the number of faces in the image from left to right is constantly increasing, so the distribution of faces is denser, and the problems of occlusion, multi-scale and density in a complex environment are fully considered, which offers convenience to fully prove the robustness and generalization ability of the model. From the analysis of the figure, it can be found that the performance of the model used in this paper is better than the other four in test results, but all the models have poor detection results for severe occlusion and half face. We consider that the cause of this problem is because of the lack of images with serious missing face features in the data set, which leads to less learning of these features and the poor generalization ability of the model. Therefore, one of the main tasks in the future is to expand the data set and enrich the diversity of features.

### 6.6. Influence of Different Activation Functions

We use the Mish activation function [42] to highlight the effect of the H-swish activation function on the results of this paper, as shown in Table 11.

It can be seen from Table 11 that in the same situation, using H-swish as the activation function can obtain better detection results. Therefore, the mask detection model has stronger nonlinear feature learning ability under the action of the H-swish activation function. At this time, the model has the highest detection accuracy in the comparison experiment of activation functions.

### 6.7. Analysis of Ablation Experiment

We use ablation experiments to analyze the influence of the improved method on the performance of the model. The experiments are divided into five groups as comparisons. The first group is YOLO-v4. In the second group, the CSP1_X module is introduced into the backbone feature extraction network module of YOLO-v4. The third group is the CSP2_X module introduced into the Neck module of YOLO-v4. In the fourth group, both CSP1_X and CSP2_X modules are added into YOLO-v4 at the same time. The last set of experiments is the result of the model in this article. The experimental results are shown in Table 12.

It can be seen from the analysis in the table that the use of the CSP1_X module in the backbone feature extraction network enhances the AP values of the three categories, and at the same time, the and FPS are increased by 2.7% and 19.64 FPS, respectively, thus demonstrating the effectiveness of CSP1_X. Different from YOLO-v4, this paper takes advantage of the CSPDarkNet53 module and introduces the CSP2_X module into Neck to further enhance the learning ability of the network in semantic information. Experiments show that CSP2_X also improves the AP values of the three categories, and mAP and FPS are increased by 2.3% and 21.62 FPS, respectively, compared with YOLO-v4. From the comparative experiments of the fourth and the fifth groups, we find that the H-swish activation function significantly ameliorates the detection accuracy of the model.

In summary, the improved strategies proposed in this paper based on YOLO-v4 are meaningful for promoting the recognition and detection of face masks in complex environments.

## 7. Conclusions

In this paper, an improved algorithm based on YOLO-v4 is proposed to solve the problem of mask wearing recognition. Meanwhile, the effectiveness and robustness of this model are verified by the comparative study of two kinds of object detection algorithms.

This article can be summarized as follows:Firstly, the CSP1_X module is introduced into the backbone feature extraction network to enhance feature extraction.Secondly, the CSP2_X module is used in the Neck module to ensure that the model can learn deeper semantic information in the process of feature fusion.Thirdly, the Hard-Swish activation function is used to improve the nonlinear feature learning ability of the model.Finally, the proposed adaptive image scaling algorithm can reduce the model’s reasoning time.

The experimental results show that the algorithm proposed in this paper has the highest detection accuracy compared with others for strict mask detection tasks. Meanwhile, the phenomena of false and missing detection have been reformed. Moreover, the algorithm in this paper effectively decreases the requirements of the model on training cost and model complexity, which enables the model to not only be deployed on medium devices but also be extended to other object detection tasks, such as mask wearing detection tasks of students, passengers, patients, and other staff.

However, in the present work, there are still some problems in insufficient feature extraction for difficult detection samples or even missing and false detection cases. In addition, the case of wearing a mask when the light is insufficient is also not considered. Therefore, the next step should be expanding the data set based on the standard mask wearing criteria and obtaining further improvements for the model in the present work, and so extending it to more object detection tasks.

## Figures and Tables

**Figure 1 sensors-21-03263-f001:**
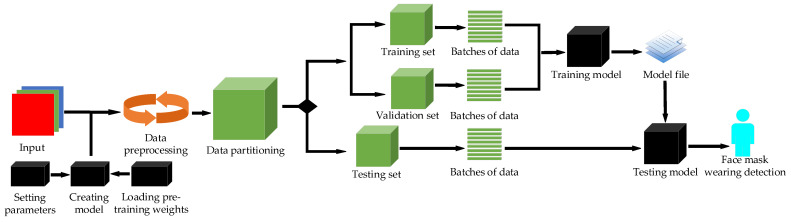
Flow chart of the proposed approach.

**Figure 2 sensors-21-03263-f002:**
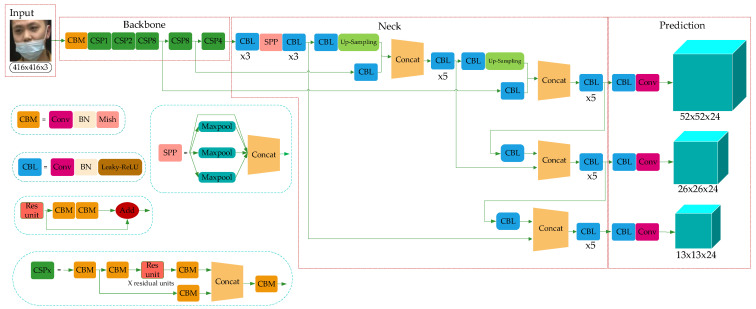
YOLO-v4 network structure.

**Figure 3 sensors-21-03263-f003:**
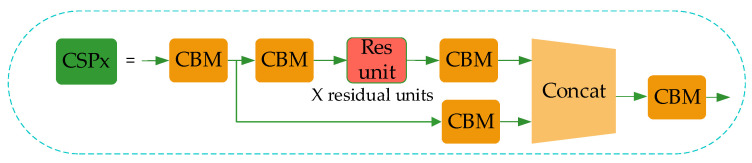
CSPDarkNet53 module structure.

**Figure 4 sensors-21-03263-f004:**
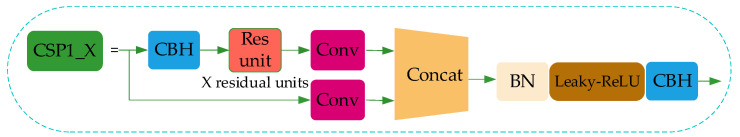
CSP1_X module structure.

**Figure 5 sensors-21-03263-f005:**
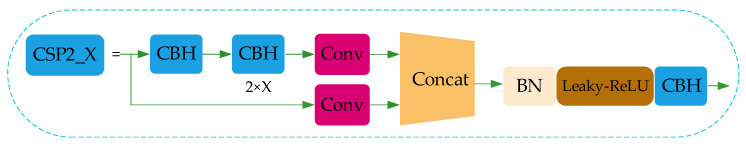
CSP2_X module structure.

**Figure 6 sensors-21-03263-f006:**
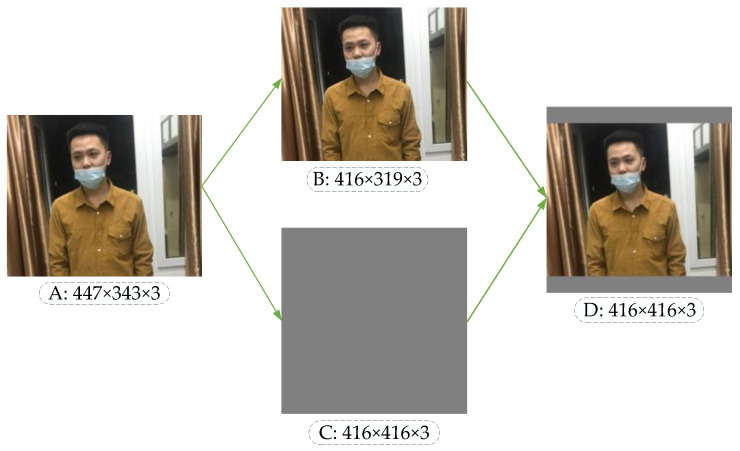
Image scaling in YOLO-v4.

**Figure 7 sensors-21-03263-f007:**
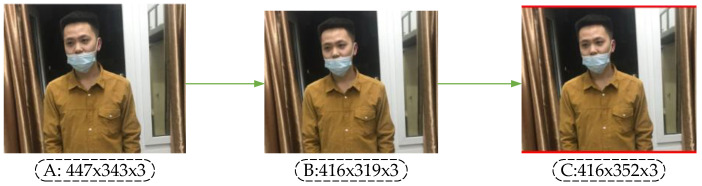
Adaptive image scaling.

**Figure 8 sensors-21-03263-f008:**
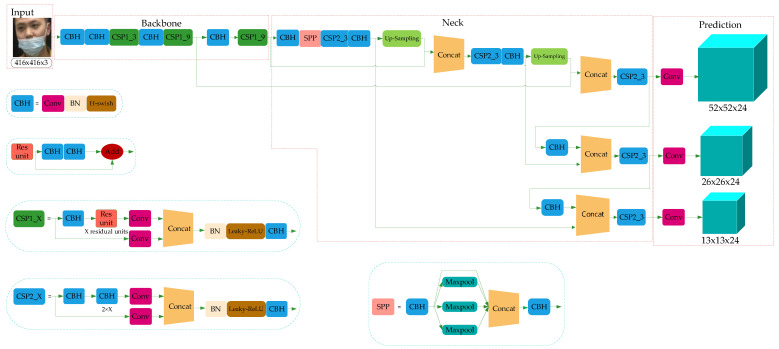
Network model of mask detection.

**Figure 9 sensors-21-03263-f009:**
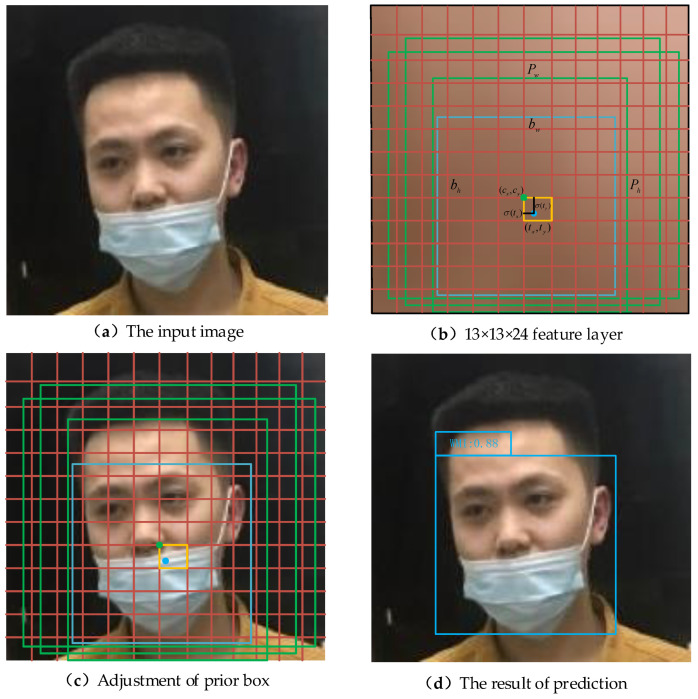
The process of object positioning and prediction.

**Figure 10 sensors-21-03263-f010:**
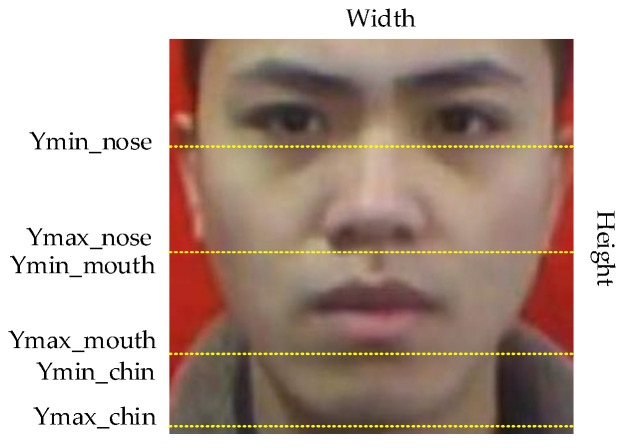
Division of key parts.

**Figure 11 sensors-21-03263-f011:**
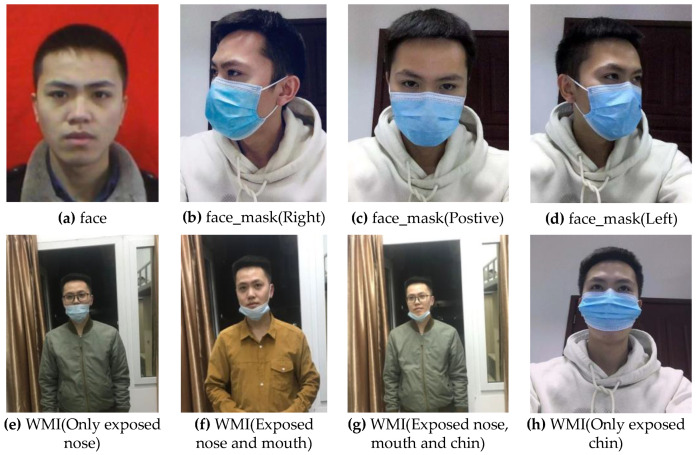
Sample diagram from the data set.

**Figure 12 sensors-21-03263-f012:**
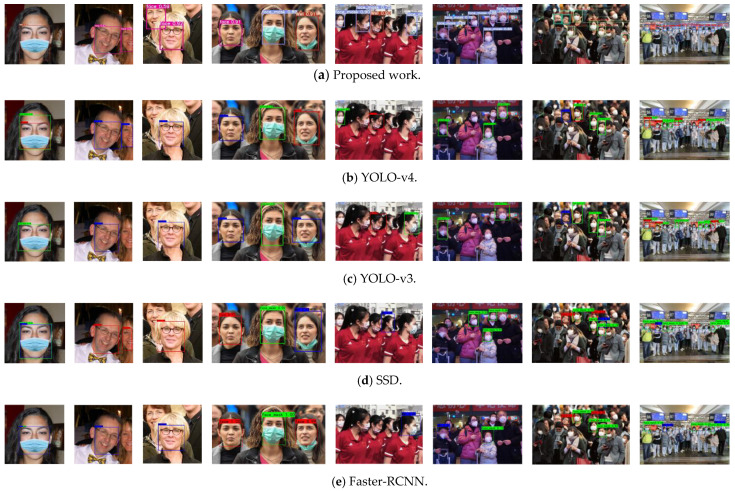
Visualization of different models in performance testing.

**Table 1 sensors-21-03263-t001:** The size of the prior box.

Feature Map	Receptive Field	Prior Box Size
13 × 13	large object	(221 × 245) (234 × 229) (245 × 251)
26 × 26	medium object	(165 × 175) (213 × 222) (217 × 195)
52 × 52	small object	(46 × 51) (82 × 100) (106 × 201)

**Table 2 sensors-21-03263-t002:** Distribution of different types of samples in the data set.

Sort	Training Set		Validation Set		Testing Set	
	Images	Objects	Images	Objects	Images	Objects
face	2556	2670	338	350	721	753
face_mask	2685	2740	219	228	716	730
WMI	2585	2604	311	311	724	730
total	7826	8014	868	889	2161	2213

**Table 3 sensors-21-03263-t003:** Configuration parameters.

Device	Configuration
Operating system	Windows 10
Processor	Inter(R)i7-9700k
GPU accelerator	CUDA 10.1, Cudnn 7.6
GPU	RTX 2070Super, 8G
Frames	Pytorch, Keras, Tensorflow
Compilers	Pycharm, Anaconda
Scripting language	Python 3.7
Camera	A4tech USB2.0 Camera

**Table 4 sensors-21-03263-t004:** The hyperparameters of the model.

Hyperparameters	Before Initialization	After Initialization
initial learning rate	0.01000	0.00320
optimizer weight decay	0.00050	0.00036
momentum	0.93700	0.84300
classification coefficient	0.50000	0.24300
object coefficient	1.00000	0.30100
hue	0.01500	0.01380
saturation	0.70000	0.66400
value	0.40000	0.46400
scale	0.50000	0.89800
shear	0.00000	0.60200
mosaic	1.00000	1.00000
mix-up	0.00000	0.24300
flip up-down	0.00000	0.00856

**Table 5 sensors-21-03263-t005:** Comparison of different models in parameters, model size, and training time.

Model	Parameters	Model Size	Training Time
Proposed work	45.2 MB	91.0 MB	2.834 h
YOLO-v4	61.1 MB	245 MB	9.730 h
YOLO-v3	58.7 MB	235 MB	8.050 h
SSD	22.9 MB	91.7 MB	3.350 h
Faster R-CNN	27.1 MB	109 MB	45.830 h

**Table 6 sensors-21-03263-t006:** Comparison of different models in test time, reasoning time, FPS.

Model	One Image Test Time	All Reasoning Time	FPS
Proposed work	0.022 s	144.7 s	54.57
YOLO-v4	0.042 s	151.1 s	23.83
YOLO-v3	0.047 s	153.1 s	21.39
SSD	0.029 s	97.0 s	34.69
Faster R-CNN	0.410 s	1620.7 s	2.44

**Table 7 sensors-21-03263-t007:** The parameter distribution of different modules in different models.

Module	Faster R-CNN	SSD	YOLO-v3	YOLO-v4	Proposed Work
Backbone	-	-	40,620,740	30,730,448	9,840,832
Neck	-	-	14,722,972	27,041,012	37,514,988
Prediction	-	-	6,243,400	6,657,945	43,080
All parameters	28,362,685	24,013,232	61,587,112	64,014,760	47,398,900
All CSPx	-	-	-	26,816,384	-
All CSP1_X	-	-	-	-	8,288,896
All CSP2_X	-	-	-	-	18,687,744
All layers	185	69	256	370	335

**Table 8 sensors-21-03263-t008:** Sample detection results of different models on the test set.

Models	Sort	Size	Object	TP	FP	FN	P	R	$F_{1}$
Proposed work	face	416 × 416	753	737	50	16	0.936	0.979	0.957
	face_mask	416 × 416	730	725	23	5	0.969	0.993	0.980
	WMI	416 × 416	730	712	39	18	0.948	0.975	0.961
	Total	416 × 416	2213	2174	112	39	0.951	0.982	0.967
YOLO-v4	face	416 × 416	753	666	42	87	0.941	0.885	0.910
	face_mask	416 × 416	730	705	199	25	0.780	0.966	0.860
	WMI	416 × 416	730	670	195	60	0.775	0.918	0.840
	Total	416 × 416	2213	2041	436	172	0.832	0.923	0.870
YOLO-v3	face	416 × 416	753	640	53	113	0.924	0.850	0.890
	face_mask	416 × 416	730	686	23	44	0.968	0.940	0.950
	WMI	416 × 416	730	623	26	107	0.960	0.853	0.900
	Total	416 × 416	2213	1949	102	264	0.950	0.881	0.913

**Table 9 sensors-21-03263-t009:** The comparative experiments of AP of different models in three categories.

Sort	Size	IOU	Face	Face_Mask	WMI
Proposed work	416 × 416	AP@.50	0.979	0.995	0.973
	416 × 416	AP@.75	0.978	0.995	0.983
	416 × 416	AP@.50:.95	0.767	0.939	0.834
YOLO-v4	416 × 416	AP@.50	0.943	0.969	0.944
	416 × 416	AP@.75	0.680	0.899	0.800
	416 × 416	AP@.50:.95	0.541	0.740	0.670
YOLO-v3	416 × 416	AP@.50	0.921	0.981	0.941
	416 × 416	AP@.75	0.617	0.888	0.835
	416 × 416	AP@.50:.95	0.559	0.789	0.724
SSD	300 × 300	AP@.50	0.941	0.986	0.988
	300 × 300	AP@.75	0.503	0.920	0.926
	300 × 300	AP@.50:.95	0.518	0.789	0.790
Faster R-CNN	600 × 600	AP@.50	0.943	0.974	0.950
	600 × 600	AP@.75	0.700	0.927	0.866
	600 × 600	AP@.50:.95	0.612	0.824	0.769

**Table 10 sensors-21-03263-t010:** The mAP comparison experiments of different models in all categories.

Model	mAP@.50	mAP@.75	mAP@.50:95
Proposed work	0.983	0.985	0.847
YOLO-v4	0.952	0.793	0.680
YOLO-v3	0.948	0.780	0.689
SSD	0.972	0.783	0.691
Faster R-CNN	0.956	0.831	0.735

**Table 11 sensors-21-03263-t011:** Influence of different activation functions.

Function	Train Time	Face	Face_Mask	WMI	mAP@.50
H-swish	2.834 h	0.979	0.995	0.973	0.983
Mish	3.902 h	0.971	0.995	0.973	0.980
L-ReLU	2.812 h	0.975	0.985	0.974	0.978
ReLU	3.056 h	0.970	0.972	0.969	0.970
Sigmoid	2.985 h	0.966	0.968	0.963	0.966

**Table 12 sensors-21-03263-t012:** Ablation experiments.

CSP1_X	CSP2_X	H-Swish	Face	Face_Mask	WMI	mAP@.50	FPS
×	×	×	0.943	0.969	0.944	0.952	23.83
√	×	×	0.982	0.984	0.972	0.979	43.47
×	√	×	0.969	0.993	0.962	0.975	45.45
√	√	×	0.971	0.993	0.967	0.977	47.65
√	√	√	0.979	0.995	0.973	0.983	54.57

## Data Availability

The data presented in this study are available on request from the corresponding author.

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
