# Peer review of "Face Mask Wearing Detection Algorithm Based on Improved YOLO-v4"

_sensors, 2021, doi:10.3390/s21093263_

Round 1
Reviewer 1 Report
The topic is interesting and the methodology is adequately described. However, the authors need to address the following comments:
- The English spelling and grammar must be significantly improved. The reviewer suggests proofreading the whole manuscript. Some examples include:
- Lines 29-30: “They can enter the lungs as humans breathe and then causing pneumonia, nerve damage or even toxic reactions, which can be so serious that endanger lives.”
- Figure 9: Incorrect spelling of the word “Mouth”
- Lines 597 to 599: “Because the objects in the data set used in this paper can be directly observed, and the wearing of face masks at night is not considered.”
- The background section is incomplete. It covers the development of object detection algorithms based on deep learning, but fails to explain the advantages of the proposed method versus other methods related to face mask detection. Further examination of the literature needs to be carried out and research gaps and study contributions need to be highlighted.
- The evaluation parameters, as stated in section 5.5, are unclear and need to be further clarified: “FP (false positive) means that other categories are detected as being related to this category by mistake; FN (false negative) indicates that the real sample is detected as the opposite result or undetected category”, the definitions of FP and FN seem to be similar.
- In Table 7, the reasoning time for Faster R-CNN is a lot higher than that of other models. However, Table 10 shows that the image size for this model (600*600) is larger than other models. Can this be the reason for the significantly slower performance of Faster R-CNN when compared with other models?
- In the Introduction section, it is stated that the aim of this paper is to develop a face mask detection method suitable for complex environments. However, according to results displayed in Figure 11, most test subjects were very close to the camera and do not actually represent a real time complex scenario. Moreover, the reviewer believes that a resolution of 316*316 is not good enough to accurately detect when the subject is far away from the camera as in a real complex environment. Do elaborate on that.
Reviewer 2 Report
When respiratory infectious diseases are prevalent or people work in dust or other polluted environments, masks play an important role in filtering viruses, poisonous, harmful gases and droplets suspended in the air. To solve the problem of low accuracy, low real-time performance, poor robustness, and other problems caused by the complex environment such as blocking, multi- scale, illumination, and density of the face in the mask detection task, this paper proposed a face mask recognition and standard wear detection algorithm based on the improved YOLO-v4 in complex environment. Firstly, an improved CSPDarkNet53 was introduced into the trunk feature extraction network, which reduced the computing cost of the network and improved the learning ability of the model. Secondly, the adaptive image scaling algorithm can reduce computation and redundancy effectively. Then, the improved PANet structure is introduced, so that the network has more semantic information in the feature layer. Finally, to standardize the wearing of masks, a face mask detection data set was made according to the standard wearing of masks, and a variety of
evaluation indexes were compared based on the object detection algorithm of deep learning to evaluate the effectiveness of the model. The topic is overall very interesting but Authors should address the following comments.
(1) In my opinion the Title of the paper is not suitable Your title is too long and should be reduced. A good title should not exceed 10 words. The title should be clear and informative and should reflect the aim and approach of the work.
Recommendations for titles:
Fewest possible words that describe the contents of the paper.
Avoid waste words like "Studies on", or "Investigations on”, “effects of”, “comparison of”, or “a case of”.
Use specific terms rather than general.
Watch your word order and syntax.
Avoid abbreviations and jargon Change Title more interesting for the readers.
(2) Abstract should be revised more technically.
(3) Major contribution is missing and Line 81~94 Authors should elaborate it more for the readers.
(4) Authors should draw the flow chart of the proposed approach in the introduction section.
(5) As this paper is related to computer aided diagnosis so authors shod ad theses paper in the introduction section.
Shin, Hoo-Chang, Holger R. Roth, Mingchen Gao, Le Lu, Ziyue Xu, Isabella Nogues, Jianhua Yao, Daniel Mollura, and Ronald M. Summers. "Deep convolutional neural networks for computer-aided detection: CNN architectures, dataset characteristics and transfer learning." IEEE transactions on medical imaging 35, no. 5 (2016): 1285-1298. Giger, Maryellen L., and Kenji Suzuki. "Computer-aided diagnosis." In Biomedical information technology, pp. 359-XXII. Academic Press, 2008./
- A. Khan and Y. Kim, "Cardiac arrhythmia disease classification using lstm deep learning approach," Computers, Materials & Continua, vol. 67, no.1, pp. 427–443, 2021. https://www.techscience.com/cmc/v67n1/41210
(6) For data imbalance authors used up sampling ?? This looks very weak contribution .
(7) On Page15 Authors missing F1 score Formula so should add F1 SCORE AND ALSO UPDATE THE Table 9 as well.
(8) Table 11. The mAP comparison experiments of different models in all categories, should be update include the Reference number as well.
(9) What is drawback of the proposed approach and future analysis??Any data complexity issue??
(10) Please conclude your manuscript in more concrete way.
(11) There are a lot of typo mistake and formatting issues, authors should proofread the updated version.
(12) All the table and equations should be align with text according to proper formatting.
Reviewer 3 Report
The paper presents an interesting subject, but the following aspects must be explained more clearly in order to increase the soundness of the paper:
- section about related work must be included, also containing results of the described methods (existing work that are already described must be added in a separate section)
- the chosen of the base network that is improved must be explained based on existing results
- the novelty of the proposed method must be explained more clearly
- uniformise references: eg. line 189: "He et al."
Round 2
Reviewer 2 Report
The authors did excellent work and resolve my previous queries very well. This paper looks interesting for the readers and I agree to accept it for publication in the present form.
Reviewer 3 Report
The title from the submitted file (Face Mask Wearing Detection Algorithm Based on Improved YOLO-v4) is slightly different from the platform title (Face Mask Recognition and Standard Wearing Detection Algorithm Based on Improved YOLO-v4 in Complex Environment).
Otherwise, since all my comment were addressed, I recommend to publish the paper.